# Optimal Design and Design Parameter Sensitivity Analyses of an eVTOL PAV in the Conceptual Design Phase

**Bong-Sul Lee, Abera Tullu and Ho-Yon Hwang ***  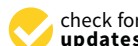

Department of Aerospace Engineering, Sejong University, 209, Neungdong-Ro, Gwangjin-Gu, Seoul 05006, Korea; 19110199@sju.ac.kr (B.-S.L.); tuab@sejong.ac.kr (A.T.)
* Correspondence: hyhwang@sejong.edu; Tel.: +82-10-6575-2282

**Abstract:** An optimization study of an electric vertical takeoff and landing personal air vehicle (eVTOL PAV) was performed during the conceptual design stage using the design of experiments method. In defining the initial problem, a design target parameter was set. The PAV subsystem was based on a configuration tradeoff study matrix, which was used to effectively conduct configuration selection. Initial sizing was performed using the PAV sizing program developed by this research team using Microsoft Excel and Visual Basic for Application (VBA). A screening test was performed to find parameters with high sensitivity among independent design parameters. The response surface method was used to model design target parameters, and a regression equation was estimated using the experimental design method. A Monte Carlo simulation was performed to confirm the feasibility of the generated model. To optimize the design independent parameter, a satisfaction function was selected, and the appropriateness of the data was determined using a Pareto plot and *p*-value.

**Keywords:** eVTOL; PAV; aircraft design; design of experiments; optimization

---

## 1. Introduction

In 2015, individuals in San Francisco spent 230 h commuting, resulting in 500,000 h of lost productivity each day. In Mumbai, the average commuting time exceeds 90 min. The more time people spend on the road, the less time they have to spend on relationships and at work, while more money is spent on fuel [1]. The number of automobiles is increasing at a steeper rate than the speed at which roads are expanding, traffic jams are intensifying, and the length of time spent on the roads is increasing, causing fossil fuel to be wasted and air pollution to grow. Because the current two-dimensional traffic system cannot effectively solve traffic jams, considerable attention is being paid to a three-dimensional traffic system, based on the use of electric vertical takeoff and landing (eVTOL) personal air vehicles (PAV).

To take off and land vertically, there is no need for a large airport, like those required for conventional fixed-wing aircraft, and because the eVTOL PAV uses electricity, there are no harmful emissions, so it has eco-friendly advantages. In addition, it has the advantage of significantly reducing congestion by using a new traffic system that is operated by designating a destination, in contrast to the existing traffic system that requires using a specific route.

Areas that are expected to first adopt this service will be those that currently do not have access to the existing roadway infrastructure. This is because the time and amount of money saved using eVTOL PAV service will be limited initially, since it is a limited transportation service. Nonetheless, it is predicted that, by 2050, 100,000 eVTOL passenger drones worldwide will be transporting humans and material resources in city centers [2].

The eVTOL PAV is getting closer to real life application, thanks to the ongoing development of technologies such as autonomous flight, communication, and electric batteries in the aviation field.

However, there are a number of obstacles to commercialization. The most prominent obstacles are safety, noise, security, infrastructure facilities, and operating costs. Collaboration between manufacturers, navigation service providers, lawmakers, and civil society is necessary to overcome these obstacles to commercialization.

In efforts to lead the new transportation paradigm, manufacturers are making eVTOL PAVs in various configurations. Typical eVTOL PAV designs include the tilt-wing type, composite type, and multi-copter type. Typical eVTOL PAVs in each category are the Vahana [3], Aurora [4], and Volopter [5]. Since the eVTOL PAV is powered by electric battery, operating range and cruising time are limited, so it is necessary to select the type of eVTOL PAV that is appropriate for the operation area and customer needs. The multicopter type has excellent vertical takeoff and landing performance and is suitable for short-distance transportation in an urban area. The tilt-wing type has excellent flight speed and fuel efficiency, making it suitable for long-distance transportation.

Bacchini and Cestino tried to understand which is the best eVTOL design, presenting and discussing all the different configurations, from the first developed in the 1950s and 1960s to the present eVTOL configurations. Then, the performances of the three main eVTOL configurations were evaluated and compared using data from existing prototypes [6]. They also compared personal air vehicle to the existing vehicles that may compete with it and addressed the estimation of its performances in hover, cruise flight, and the transition phase. The main parameters affecting performances are then discussed. Considerable space is dedicated to the battery mass to total mass ratio [7].

Kadhiresan and Duffy gave special attention to the tradeoff between configuration classes with regard to efficiency and suitability for different missions [8]. They used build-up component-based weight models to size several types of configurations for varying combinations of cruise range and cruise speed, which are two of the most important variables for defining the utility of an aircraft as an on-demand mobility (ODM)/urban air mobility (UAM) vehicle. With this information, they then determined the set of mission profiles for which each configuration dominates performance and efficiency. Holding aircraft footprint constant, they also demonstrated design sensitivity to sweeps over rotor diameter, wing loading, and battery energy density, and applied these trade studies to enhancing the aircraft's design and performance.

Unlike existing aircraft, there is still limited data on eVTOL PAV performance, so it is not easy to design the optimal aircraft. Because there is not enough data, manufacturers are having difficulty in designing and experimenting with various designs of eVTOL PAVs. To efficiently design an eVTOL PAV, it is necessary to employ various optimization techniques from the conceptual design stage [9].

## 2. eVTOL PAV Type

VTOL aircraft do not require runways for vertical takeoff and landing. A rotorcraft, such as a helicopter, is capable of vertical takeoff and landing, but has the disadvantage of being noisy and unable to fly at high speed. A fixed-wing aircraft has the advantage of being able to fly at high speed but has the disadvantage of requiring a runway for takeoff. Therefore, the optimal design for VTOL is an aircraft that can take off and land vertically and also fly at high speed.

In the early days, VTOL aircraft used internal combustion engines, but designers began using electric motors to avoid the high noise and environmental problems caused by fossil fuel engines. Current eVTOL aircraft can be divided into a multicopter type, a composite type, a tilt wing type, a tilt rotor type, and a tilt ducted fan type, depending on the propulsion method. Table 1 summarizes representative eVTOL PAVs for each type.

The most effective eVTOL PAV is different for each given mission profile. The multicopter type is excellent for hovering performance, but it is only suitable for short-range flight because it is relatively slow and has inefficient energy consumption when cruising. The tilt rotor and tilt wing types have problems such as instability in control during transition from hovering to forward flight, but they are suitable for long-distance flight because their speed is relatively fast and energy efficient when cruising. The complex type has intermediate performance, between that of the multicopter type and the tilt rotor

type. The performances of the tilted wing type Vahana, the complex type Aurora, and the multicopter type Volocopter are shown in Table 2, and the required time by mission range is shown in Figure 1.

**Table 1.** Electric vertical takeoff and landing personal air vehicle (eVTOL PAV) type.

| PAV Type | Figure |
| --- | --- |
| Tilt wing [3] |  |
| Complex [4] |  |
| Multicopter [5] |  |
| Tilt rotor [10] |  |
| Tilt ducted fan [11] |  |

**Table 2.** eVTOL PAV specification.

| Category | Vahana | Aurora | Volocopter | Unit |
|---|---|---|---|---|
| Seats | 1 | 2 | 1 | PAX |
| Fuselage length | 5.7 | 9.14 | 5.25 | m |
| Overall height | 2.81 | 3.35 | 2.5 | m |
| Wing span | 6.25 | 14 | - | m |
| Empty weight | 475 | 565 | 700 | kg |
| MTOW | 815 | 800 | 900 | kg |
| Useful load | 90 | 225 | 200 | kg |
| Range | 50 | 80 | 35 | km |
| Altitude | | 1524 | | m |
| Cruise speed | 200 | 180 | 100 | km/h |
| Motor output | $8 \times 45$ | $8 \times 75$ | $4 \times 140$ | kW |

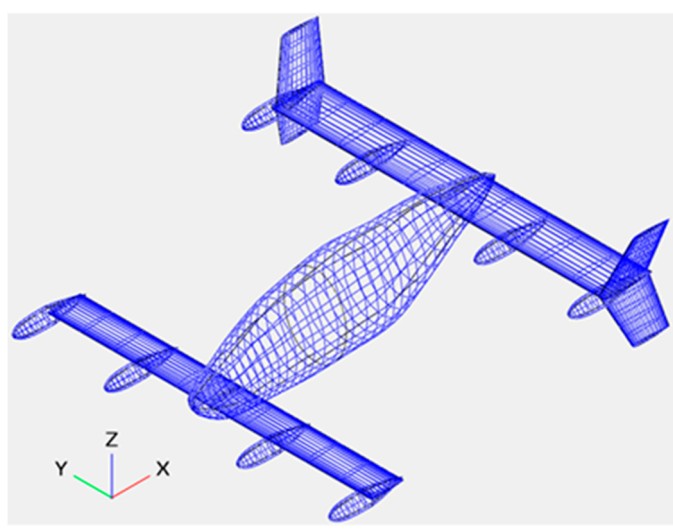

**Figure 1.** Vahana configuration using OpenVSP.

Since the tilt wing type configuration shows better cruise speed and range than other configurations, the Vahana was chosen as a baseline model of this study. The performance of the Vahana battery was assumed to be the same as the Tesla 3 model. Tables 3 and 4 shows the specifications of the 2170 battery and module, which is made in a modular way so the battery cell can be used in the aircraft.

**Table 3.** Specifications of the 2170 battery cell.

| Category | Specification | Unit |
|---|---|---|
| Weight | 70 | g |
| Volume | 970 | $mm^3$ |
| Capacity | 17.3 | Wh |
| Density | 247 | Wh/kg |

**Table 4.** Specifications of the 2170 battery cell module.

| Category | Specification | Unit |
|---|---|---|
| Width | 292 | mm |
| Thick | 90 | mm |
| Length | 1715 | mm |
| Weight | 86.6 | kg |

The energy required for hovering during takeoff and landing is based on the momentum theory [12].

$$P = \sqrt{\frac{T^3}{2\rho_{sea}A_{hover}}} \tag{1}$$

where $T$ is the hovering thrust, $\rho_{sea}$ is the air density at sea level, and $A_{hover}$ is the disk actuator area.

Using Equation (1), Vahana's required hovering power and maximum hovering time are shown in Table 5.

**Table 5.** Maximum hovering time for Vahana.

| Category | Vahana | Unit |
|---|---|---|
| Power required | 56.85 | kW |
| Hovering time | 0.67 | h |

The electric energy consumption of the aircraft is [13]

$$E^* m_{batt} \eta_t = W\frac{D}{L}R \tag{2}$$

where $E^*$ is the battery energy density, $m_{batt}$ is the battery mass, $\eta_t$ is the consumption efficiency, $W$ is the mass, $L.$ is the lift force, $D$ is the drag force and $R$ is the range.

Rewriting the equation for range, it can be expressed as

$$R = 0.75E^* \frac{m_{batt}}{m}\frac{1}{g}\frac{L}{D} \tag{3}$$

where $m$ is the aircraft mass, $g$ is the gravity acceleration and $\eta_t$ is assumed to be 0.75.

In order to calculate the flight range in Equation (3), a drag force is calculated using Equations (4) and (5) [14].

$$D = \frac{1}{2}\rho v^2 S C_D \tag{4}$$

$$C_D = C_{D_v} + C_{D_i} \tag{5}$$

where $\rho$ is the air density, $v$ is the aircraft velocity, $S$ is the wing area, $C_{D_v}$ is the parasite drag, and $C_{D_i}$ is the induced drag.

The parasite drag coefficient value is expressed as the following equation, assuming an ideal elliptical lift distribution [14].

$$C_{D_i} = \frac{C_L{}^2}{\pi e AR} \tag{6}$$

$$AR = \frac{b^2}{S} \tag{7}$$

where $C_L$ is the lift coefficient, $e$ is the Oswald span efficiency coefficient, $AR$ is the aspect ratio, and $b$ is the wing span.

The lift coefficient is as follows when cruising [14].

$$C_L = \frac{W}{\frac{1}{2}\rho v^2 S} \tag{8}$$

$$e = 1.78\left(1 - 0.045AR^{0.68}\right) - 0.64 \tag{9}$$

Replacing $C_L$ defined in Equation (8) into Equation (6), the induced drag coefficient is expressed by the following equation.

$$C_{D_i} = \frac{1}{\pi eAR}\left(\frac{W}{\frac{1}{2}\rho v^2 S}\right)^2 \tag{10}$$

Replacing $C_{D_i}$ defined in Equation (10) into Equation (5), Equation (5) is represented by the following equation.

$$C_D = C_{D_v} + \frac{1}{\pi eAR}\left(\frac{W}{\frac{1}{2}\rho v^2 S}\right)^2 \tag{11}$$

Replacing $C_D$ defined in Equation (11) into Equation (4), Equation (4) can be expressed by the following equation.

$$D = \frac{1}{2}\rho v^2 S\left(C_{D_v} + \frac{1}{\pi eAR}\left(\frac{W}{\frac{1}{2}\rho v^2 S}\right)^2\right) \tag{12}$$

In order to calculate the total drag, it is necessary to calculate the parasite drag. The component build up method was used to calculate the parasite drag. The parasite drag at subsonic speed is expressed by the following equation [14].

$$C_{D_v} = \frac{\sum\left(C_f FFQ S_{wet}\right)}{S_{ref}} + C_{D_{misc}} + C_{D_{L\&P}} \tag{13}$$

where $C_f$ is the surface friction coefficient, $FF$ is the form factor, $Q$ is the interference coefficient, $S_{wet}$ is the wetted area, $S_{ref}$ is the reference area, $C_{D_{misc}}$ is the miscellaneous drag, and $C_{D_{L\&P}}$ is the leakages and protuberances drag coefficient.

The surface friction coefficient of the plate is expressed by the following equation for laminar and turbulent flows [14].

Laminar:

$$C_f = \frac{1.328}{\sqrt{R}} \tag{14}$$

Turbulent:

$$C_f = \frac{0.455}{(log_{10}R)^{2.58}(1 + 0.144M^2)^{0.65}} \tag{15}$$

where $R$ is the Reynolds number and $M$ is the Mach number.

The form factor is represented by the following formula for each form [14].

Wing, tail wing, strut, and pylon:

$$FF = [1 + \frac{0.6}{\left(\frac{x}{c}\right)_m}\left(\frac{t}{c}\right) + 100\left(\frac{t}{c}\right)^4][1.34M^{0.18}(\cos \Lambda_m)^{0.28}] \tag{16}$$

Fuselage and canopy:

$$FF = 1 + (0.35/f) \tag{17}$$

Nacelle and smooth external store:

$$FF = (1 + \frac{60}{f^3} + \frac{f}{400}) \tag{18}$$

$$f = \frac{l}{d} = \frac{l}{\sqrt{(4/\pi)A_{max}}} \tag{19}$$

where $(x/c)_m$ is the chord-wise location of the airfoil maximum thickness point, $\frac{t}{c}$ is the airfoil thickness ratio, $\Lambda_m$ is the sweep of the maximum thickness line, $l$ is the characteristic length, and $A_{max}$ is the maximum cross section area.

The miscellaneous drag coefficient was assumed to be 5% of the parasite drag, and 20% was added if there was a skid-type landing gear. The leakage and protrusion drag coefficient was assumed to be 7.5% of parasite drag [14].

The values required to obtain the parasite drag of Vahana were obtained using OpenVSP (v3.18.0, released on 3 September 2019. NASA Langley Research Center, VA, USA) [15] and XFLR5 (v6.47, released on 7 July 2019. Open Source VLM Software) [16] and are shown in Figure 1 and Tables 6–9. The parasite drag of Vahana were obtained using Equations (12)–(18), and are shown in Table 10.

**Table 6.** Flat-plate skin friction coefficient of Vahana.

| Category | | Laminar [Count] | Turbulent [Count] | Re [-] | Mach [-] |
|---|---|---|---|---|---|
| **Vahana** | Fuselage | - | 26.9 | 19,879,625 | 0.166 |
| | Main wing | 1.1 | 30.8 | 3,138,330 | |
| | Canard wing | 1.3 | 33.1 | 2,120,493 | |
| | Nacelle | - | 33.8 | 4,771,110 | |

**Table 7.** Form factor of Vahana.

| Category | | FF | f | Max Thickness Angle | Max Thickness Point | Thickness Ratio |
|---|---|---|---|---|---|---|
| **Vahana** | Fuselage | 1.9 | 4.1 | - | - | - |
| | Main wing (e1230) | 1.4 | - | 0 | 0.309 | 0.174 |
| | Canard wing (Roncz1145MS) | 1.2 | - | 0 | 0.397 | 0.137 |
| | Nacelle | 1.0 | 10 | - | - | - |

**Table 8.** Interference coefficient of Vahana.

| Category | | Q (Interference Coefficient) |
|---|---|---|
| **Vahana** | Fuselage | 1 |
| | Main wing | 1 |
| | Canard wing | 1 |
| | Inner nacelle | 1.3 |
| | Outer nacelle | 1 |

**Table 9.** Vahana wetted area.

| Category | | Wetted Area | Unit |
|---|---|---|---|
| **Vahana** | Fuselage | 17.89 | m$^2$ |
| | Main wing | 10.07 | |
| | Canard wing | 6.11 | |
| | Nacelle | 0.77 | |

The Distributed Electric Propulsion (DEP) system spreads thrust around the aircraft, either by using three or more smaller electric propulsion units. Recent analytic and experimental distributed electric propulsion studies suggest several improvements in aircraft performance. They include energy consumption efficiency, noise abatement, steep climbing for short takeoff and landing (STOL), novel control approaches (in particular eliminating control surfaces for roll, pitch and yaw moments),

and high bypass ratios. DEP system will increase the air speed above the wing at low speed so that it will increase lift and require smaller wing area compared to the clean wing. However, OpenVSP and XFLR5 cannot handle DEP effects, so they are not considered in this study.

**Table 10.** Vahana parasite drag.

| Category | Vahana | Unit |
|:---:|:---:|:---:|
| $C_{D_s}$ | 340 | |
| $C_{D_{misc}}$ | 119 | |
| $C_{D_{L\&P}}$ | 26 | count |
| $C_{D_P}$ | 485 | |

The induced drag is calculated using Equations (20)–(22) [14].

$$C_{D_i} = KC_L{}^2 \tag{20}$$

$$K = 1/(\pi e AR) \tag{21}$$

$$e = 1.78(1 - 0.045AR^{0.68}) - 0.64, \ (\Lambda_{LE} \ \leq \ 30°) \tag{22}$$

where $\Lambda_{LE}$ is the sweep back angle of the leading edge.

The $C_L$ value was obtained by referencing Vahana's wing using Open VSP and XFLR5. The configuration of Vahana is shown in Figures 1–3 and $C_L$ value is shown in Figures 4 and 5. Using Equations (20) through (22), the induced drag of Vahana is shown in Table 11.

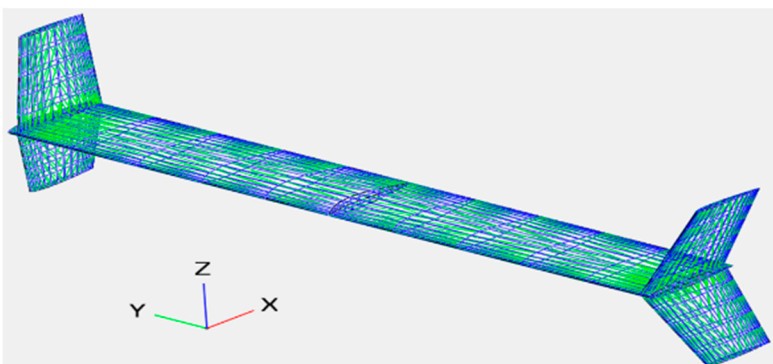

**Figure 2.** Vahana main wing by using OpenVSP.

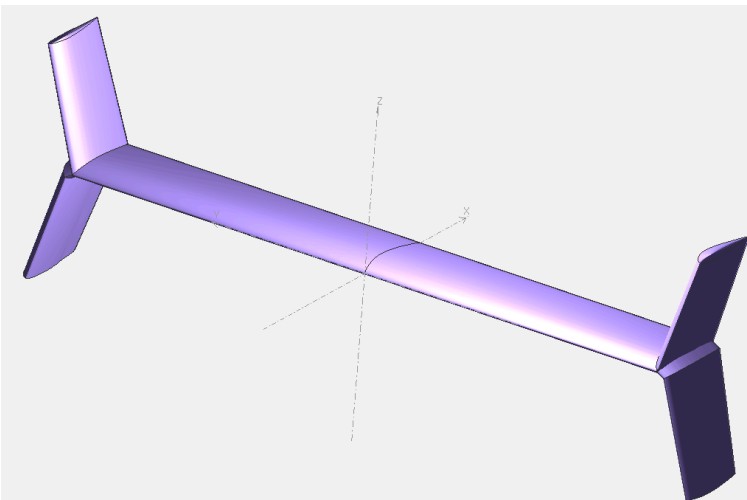

**Figure 3.** Main wing configuration of Vahana by using XFLR5.

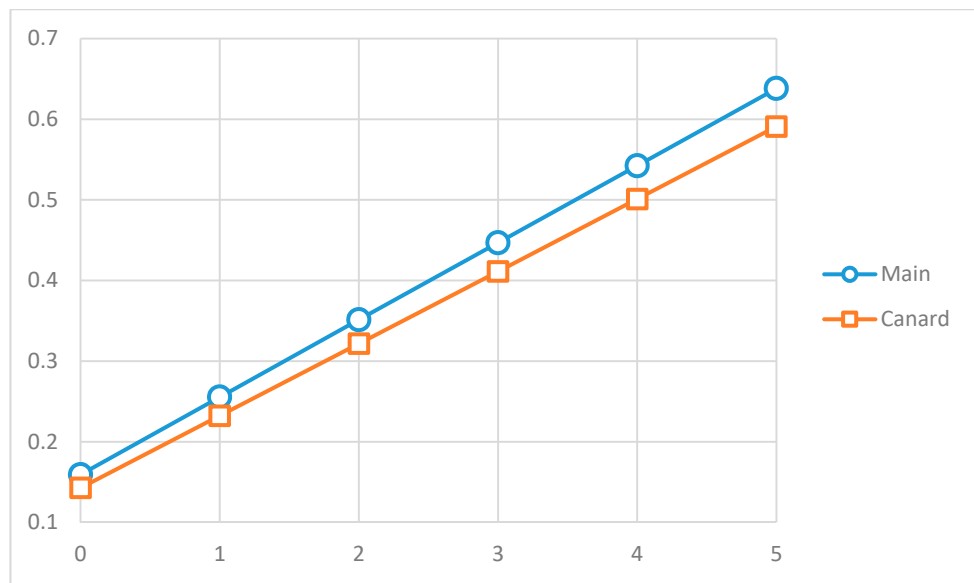

**Figure 4.** Vahana CL calculated using OpenVSP.

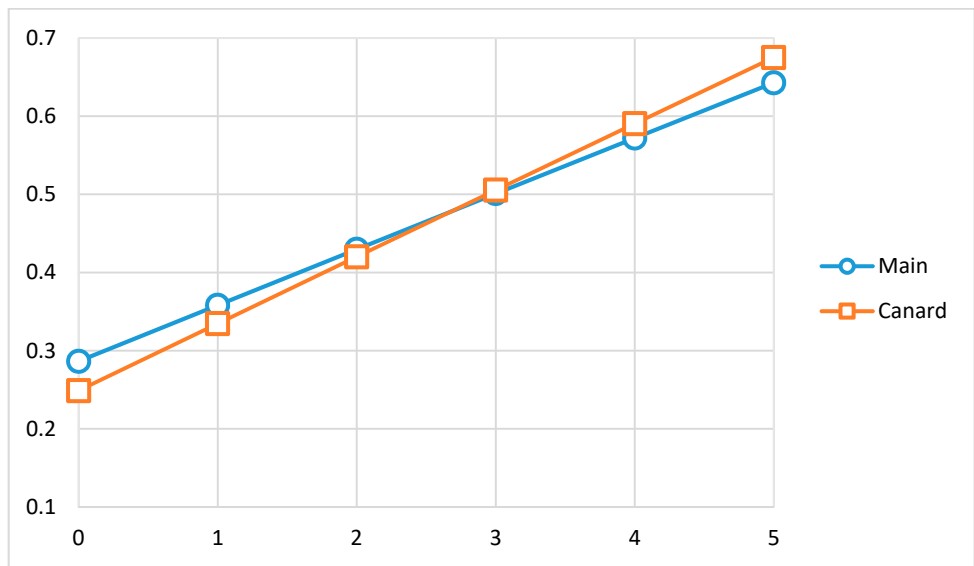

**Figure 5.** Vahana $C_L$ calculated using XFLR5.

**Table 11.** Vahana induced drag.

| $AOA=3°$ | Vahana | | Unit |
|---|---|---|---|
| | **Main Wing** | **Canard Wing** | |
| **Airfoil** | e1230 | Roncz1145MS | - |
| $C_L$ | 0.45 | 0.41 | - |
| AR | 6.96 | 9.95 | - |
| e | 0.84 | 0.76 | - |
| K | 0.05 | 0.04 | - |
| $C_{D_i}$ | 109 | 71 | count |

Table 12 shows the total drag and range obtained using the previously calculated parasite drag and induced drag.

**Table 12.** Drag and range of Vahana.

| Category | Vahana | Unit |
| --- | --- | --- |
| $C_{D_v}$ | 485 | count |
| $C_{D_i}$ | 180 | count |
| D | 1130 | N |
| Range | 188 | km |

The mission profile was set as shown in Figure 6 [17]. Takeoff and landing with hovering, climb, and forward flight are shown in section B, and descend and forward flight in section D. According to the FAA, it is illegal to operate within 1000 ft above fully populated areas and downtown areas, so takeoff was vertically up to 1000 ft and then proceeded with the climb and forward flight at the same time [18]. After climbing to 1500 ft, the vehicle starts cruise flight. After cruise is complete, the descent and forward flight will be carried out at the same time, and at the point of arrival, the aircraft will land vertically without forward flight. The total travel times were calculated for 4 km, 17 km, 35 km, and 50 km mission profiles and are shown in Figure 7. The total travel time for each mission range was calculated by dividing mission ranges by speed given in Table 2.

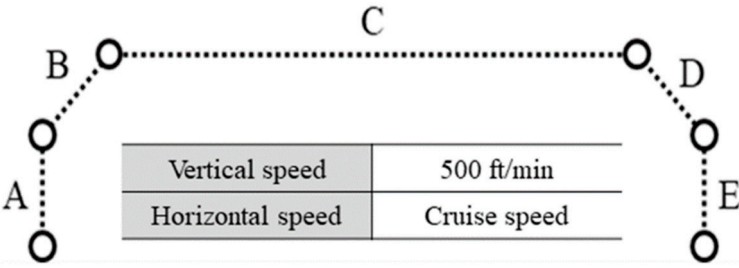

**Figure 6.** Mission profile and speed.

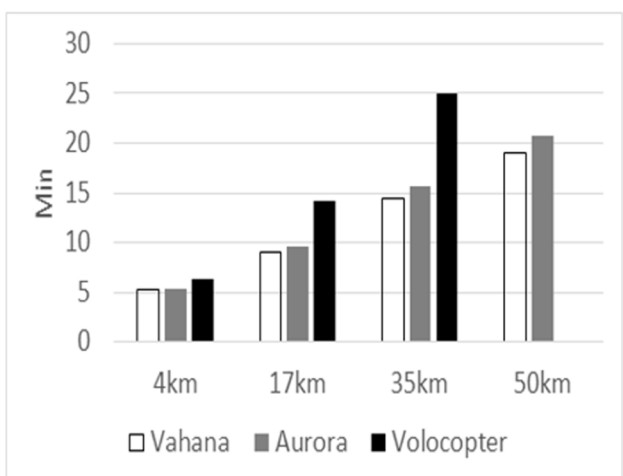

**Figure 7.** Total travel time for each mission range.

## 3. Design of Experiment

The design of experiments (DOE) is a planning method for experiments, and it explains how experiments on the problem to be solved will be conducted and how to handle the data. It can be defined as planning to get the maximum information from the minimum number of experiments when analyzing the data with any statistical method. The goal of the DOE is to find the most economical condition by selecting various factors and levels affecting the characteristics of the product, conducting experiments to find out the relationship between them, and obtaining the resulting data and analyzing it with appropriate statistical techniques.

### 3.1. Design of Experiment Steps

In general, the procedure for a DOE is explained in Figure 8.

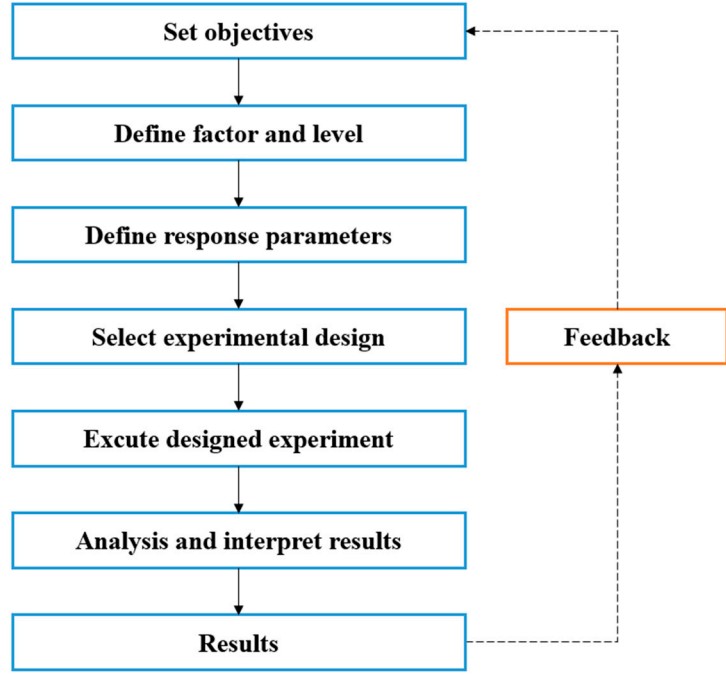

**Figure 8.** Design of experiments (DOE) flow chart.

Interpretation of experimental results should only be made within the conditions given in the experiment. Beyond this range, no conclusion can be drawn.

When the optimum conditions are obtained from the experimental results, it is necessary to estimate the characteristic values under these conditions and then to conduct a verification experiment to confirm that the optimal conditions obtained are appropriate. Experiments must be reproducible, which means repeatable, when designing experiments.

### 3.1.1. Set Objectives

In the problem definition step, the problem is to define the design requirements of the eVTOL PAV, and set new design goals according to design requirements. The target design parameters are shown in Table 13.

**Table 13.** Objective design parameters and objective values.

| Design Parameter | Target Value |
|---|---|
| MTOW | ≤2200 [lb] |
| P/W | ≤0.27 [hp/lb] |
| Width | ≤50 [ft] |
| L/D cruise | ≥12 |

The maximum takeoff weight (MTOW) is an important parameter that has a direct effect on the overall flight performance of the aircraft, such as range, cruise speed and stall speed. The design target parameter of the MTOW was set by referring to the tilt wing type eVTOL PAV, Vahana, which has the lowest fuel consumption rate when cruising.

The horsepower to weight ratio (P/W) is one of the important factors affecting the maneuverability of the aircraft. In particular, it is an important factor influencing engine and aircraft sizing. An eVTOL PAV cannot use the P/W of conventional aircraft because it requires a large P/W during takeoff and

landing. In addition, it is difficult to obtain statistical data about tilt-wing eVTOL PAVs due to the lack of open data. This study used the statistical data of a tilt-rotor aircraft to design the P/W parameter.

The aircraft size should be designed to fit the size of the landing site. Size limitations for eVTOL PAVs were published by Uber [1,17].

The lift to drag ratio (L/D) is expressed as the ratio of lift and drag, and represents the overall aerodynamic performance of the aircraft. The L/D design target parameter is based on the eVTOL PAV data given by Uber [1].

### 3.1.2. Define Factors and Response Parameters

Each eVTOL PAV has geometric design elements, such as the number of seats, distributed electric propulsion (DEP), wing position, winglet shape, tail wing shape, and landing gear shape. This means many subsystem components options should be analyzed before adoption. A tradeoff study for each component should be performed and compared with all design candidates. However, the design cost will increase exponentially if many subsystem combinations are considered. Therefore, the tradeoff study should be performed for a selected few configurations.

In order to effectively perform a series of processes, it is necessary to construct a configuration tradeoff study matrix. The configuration tradeoff study matrix organizes the combinations of subsystems that make up the aircraft in the form of a table, to determine alternatives by deriving the design configuration of the aircraft that is feasible within the range of available design techniques. The configuration tradeoff study matrix is shown in Figure 9.

| Category | | Tilt wing | Tilt rotor | Multi copter | Compounded |
|---|---|---|---|---|---|
| Seats | One | ▨ | | ▨ | |
| | Two | | | | ▨ |
| | Three | | | | |
| | Four | | ▨ | | |
| DEP (Number of propulsion units) | A(3) | | | | |
| | B(4~6) | | ▨ | | |
| | C(7~10) | ▨ | | | ▨ |
| | D(11~20) | | | ▨ | |
| | E(20~ ) | | | | |
| Position of wing | High | ▨ | ▨ | | |
| | Middle | | | | ▨ |
| | Low | | | | |
| Winglet | Rounded | | | | |
| | Sharp | | | | |
| | Cut-off | | | | |
| | End plate | | | | ▨ |
| | Split-tip | ▨ | | | |
| Tail wing | Conventional | | | | |
| | T-tail | | | | |
| | H-tail | | | | ▨ |
| | V-tail | | ▨ | | |
| | Y-tail | | | | |
| Landing gear | Conventional | | | | |
| | Bicycle | | ▨ | | |
| | Single main gear | | | | |
| | Skid | ▨ | | ▨ | ▨ |

**Figure 9.** Configuration tradeoff study matrix.

### 3.1.3. Define Design Space

After the main configuration of the aircraft is determined, the design parameters that may affect the eVTOL PAV performance physically should be specified. Because design parameters are not isolated during the conceptual design process but vary continuously, as shown in Table 14, the objective is to find optimal conditions for the design by repeating the calculation process for particular regions.

**Table 14.** Design space of design parameters.

| Design Parameter | Min. Value | Max. Value | Standard Value | Unit |
|---|---|---|---|---|
| Range | 40 | 50 | 45 | mile |
| Max speed | 110 | 130 | 120 | kt |
| Cruise speed | 100 | 120 | 110 | kt |
| Cruise altitude | 5000 | 10,000 | 7500 | ft |
| Passengers | 1 | 3 | 2 | PAX |
| Baggage | 40 | 120 | 80 | lb |
| Rate of climb | 500 | 1000 | 750 | ft/min |
| Stall speed | 36 | 40 | 38 | kt |
| Service ceiling | 10,000 | 15,000 | 12,500 | ft |
| Turn speed | 60 | 90 | 75 | kt |

### 3.1.4. Select Experimental Designs

In order to evaluate the effects of the concepts chosen from the tradeoff study matrix, modeling and simulation are necessary. This requires a clear understanding of the performance of the baseline case. In this study, as a tool for modeling and simulation, the software programs of MS Excel, JMP developed by SAS [19], and the aforementioned PAV sizing tool developed for this study were employed.

- PAV sizing program

The PAV sizing program developed by the research team was programmed with Excel and Visual Basic for Application (VBA), and the initial sizing was performed by selecting the subsystems and mission profile. The PAV's subsystems include the fuselage material, the configuration of the main wing and winglet, the propulsion method, the number of propeller blades, and the configuration of the landing gear. After selecting the design parameters, the subsystem, and mission profile of the eVTOL PAV, the optimized eVTOL PAV sizing result can be obtained according to the options (1. Minimizing the required power/2. Minimizing the wing area).

The program is composed of the mission sheet, concepts sheet, technology sheet, constraint analysis sheet, aerodynamics sheet, propulsion sheet, and weight sheet. The sizing results that can be obtained are the weight of the aircraft, MTOW, empty weight, and pay load, and the wing area, root chord length, and tip chord length of the main and tail wings. The PAV sizing program flow chart is shown in Figure 10.

Screening tests can be carried out based on the modeling and simulation to find independent design parameters that have a significant impact on design target parameters. These design independent parameters can be quantitatively informed of their relationship to the design target parameters, and can identify which design target parameters are sensitive to which design independent parameters by selecting highly sensitive design independent parameters from many design independent parameters. This enables them to be used for modeling the eVTOL PAVs. In addition, Monte Carlo simulations can be applied after modeling to find possible design ranges for target design parameters.

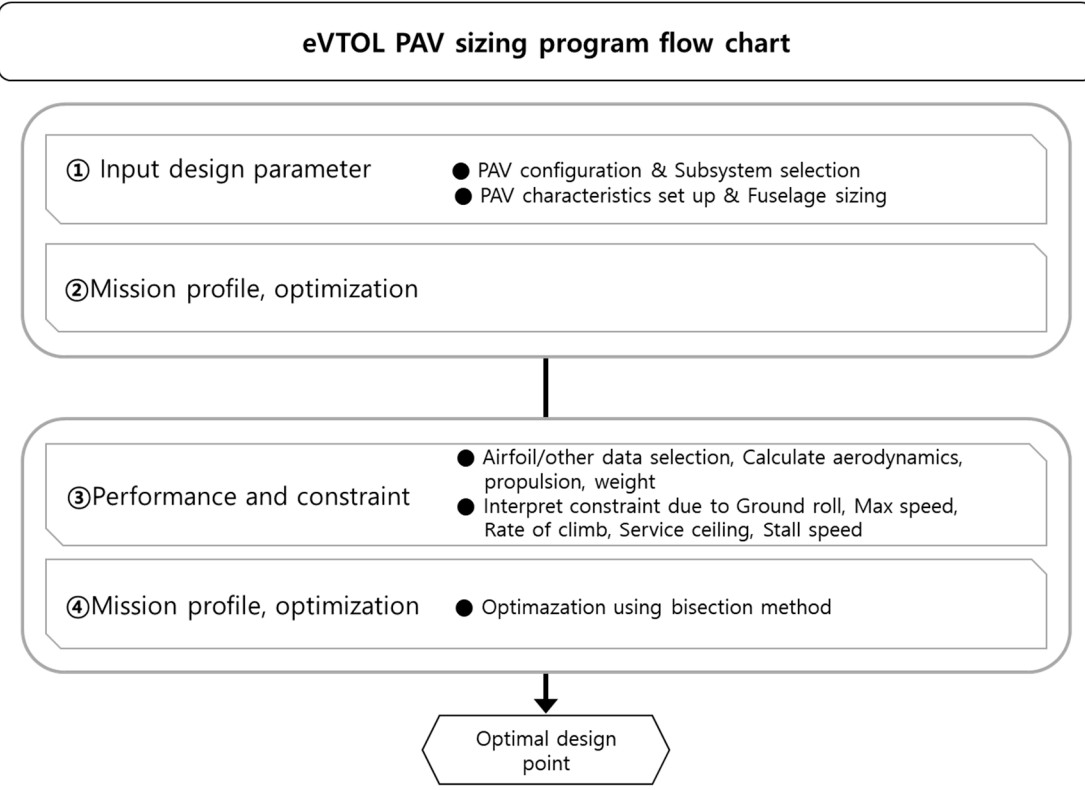

**Figure 10.** PAV sizing program flow chart.

- Screening test

Screening testing is one of the methods used to find highly sensitive design independent parameters for design target parameters before using the response surface method. During the design of experiment, fractional factorial design was used to reduce the number of experiments. Assuming that there are 10 factors with two levels in the full factorial design, the total number of experiments is $2^{10} = 1024$. The number of experiments increases exponentially depending on the level and number of factors. The experiment was carried out efficiently by reducing the number of experiments, by using the partial factorial design method, which obtains the necessary data during experiments.

- Response surface method

Modeling uses the response surface method to estimate the regression equation for each design target parameter. The response surface method was performed using the central composite design method. In a factorial design with three or more factors, the number of experiments increases exponentially, which is expensive and inefficient. However, the central composite design method can be efficiently modeled because a secondary model can be designed by adding additional axial points and central points [20].

- Monte Carlo simulation

Monte Carlo simulation is a probability calculation using random numbers. A deterministic model with a certain relationship between parameters can accurately find the predicted value, but a stochastic model cannot accurately predict a result because the relationship of the parameter is not clear. In general, a definitive model can find an analytical solution, but the probability model is often impossible. Therefore, the probability model finds the predicted value of the model by generating random numbers.

- Desirability function

The desirability function is used to optimize multiple design parameters. The desirability function is used to define the functions as smaller and larger. After setting the characteristics of the design parameters such as smaller, larger the better, optimization is performed with the geometric mean of the defined desirability functions [21].

### 3.1.5. Execute Design of Experiment and Analysis Data

- Screening test

In the screening test, 10 independent design parameters were selected as two factor levels that distinguish only the maximum and minimum values, and the test was constructed with the fractional factorial design of experiments.

In the fractional factorial design of experiments, a total of 65 experiments were conducted. Using the JMP statistical program [19], a fractional factorial design of experiments was conducted. The fractional factorial design of experiments is expressed by dividing the factor into two levels, the maximum value and the minimum value denoted by 1 and −1.

In order to use the fractional factorial design of experiments method using the JMP statistical program, the values expressed as 1 and −1 must be converted into actual values. Therefore, the maximum and minimum values of the independent design parameters were converted to fit the partial factor notation, and the corresponding data are shown in Figure 11.

| Pattern | Range | Maximum speed | Cruising speed | Cruising altitude | Passengers | → | Range | Maximum speed | Cruising speed | Cruising altitude | Passengers |
|---|---|---|---|---|---|---|---|---|---|---|---|
| -------+++ | -1 | -1 | -1 | -1 | -1 | | 40 | 110 | 100 | 5000 | 1 |
| -----++---- | -1 | -1 | -1 | -1 | -1 | | 40 | 110 | 100 | 5000 | 1 |
| ----+-+--+ | -1 | -1 | -1 | -1 | 1 | | 40 | 110 | 100 | 5000 | 3 |
| ----++-++- | -1 | -1 | -1 | -1 | 1 | | 40 | 110 | 100 | 5000 | 3 |
| ---+--+++- | -1 | -1 | -1 | 1 | -1 | | 40 | 110 | 100 | 10000 | 1 |
| ---+-+---+ | -1 | -1 | -1 | 1 | -1 | | 40 | 110 | 100 | 10000 | 1 |
| ---++----- | -1 | -1 | -1 | 1 | 1 | | 40 | 110 | 100 | 10000 | 3 |
| ---+++++++ | -1 | -1 | -1 | 1 | 1 | | 40 | 110 | 100 | 10000 | 3 |
| --+-------+ | -1 | -1 | 1 | -1 | -1 | | 40 | 110 | 120 | 5000 | 1 |
| --+--++++- | -1 | -1 | 1 | -1 | -1 | | 40 | 110 | 120 | 5000 | 1 |
| --+-+-++++ | -1 | -1 | 1 | -1 | 1 | | 40 | 110 | 120 | 5000 | 3 |
| --+-++---- | -1 | -1 | 1 | -1 | 1 | | 40 | 110 | 120 | 5000 | 3 |
| --++--+--- | -1 | -1 | 1 | 1 | -1 | | 40 | 110 | 120 | 10000 | 1 |
| --++-+-+++ | -1 | -1 | 1 | 1 | -1 | | 40 | 110 | 120 | 10000 | 1 |
| --+++--++- | -1 | -1 | 1 | 1 | 1 | | 40 | 110 | 120 | 10000 | 3 |
| --++++++--+ | -1 | -1 | 1 | 1 | 1 | | 40 | 110 | 120 | 10000 | 3 |
| -+-------+- | -1 | 1 | -1 | -1 | -1 | | 40 | 130 | 100 | 5000 | 1 |

**Figure 11.** Converting fractional values to design parameter values.

The suitability of the model obtained through the screening test can be judged using the graph in Figure 12—the narrower it is, the better the modeling. After the regression analysis, a Pareto plot and a *p*-Value were used to classify design independent parameters that have a great influence on design target parameters using the fractional factorial design of experiments.

The Pareto graph shown in Figure 13 illustrates which factors affect the target design parameters more. The factors distributed over the upper side are factors that have a greater influence on the target design parameter.

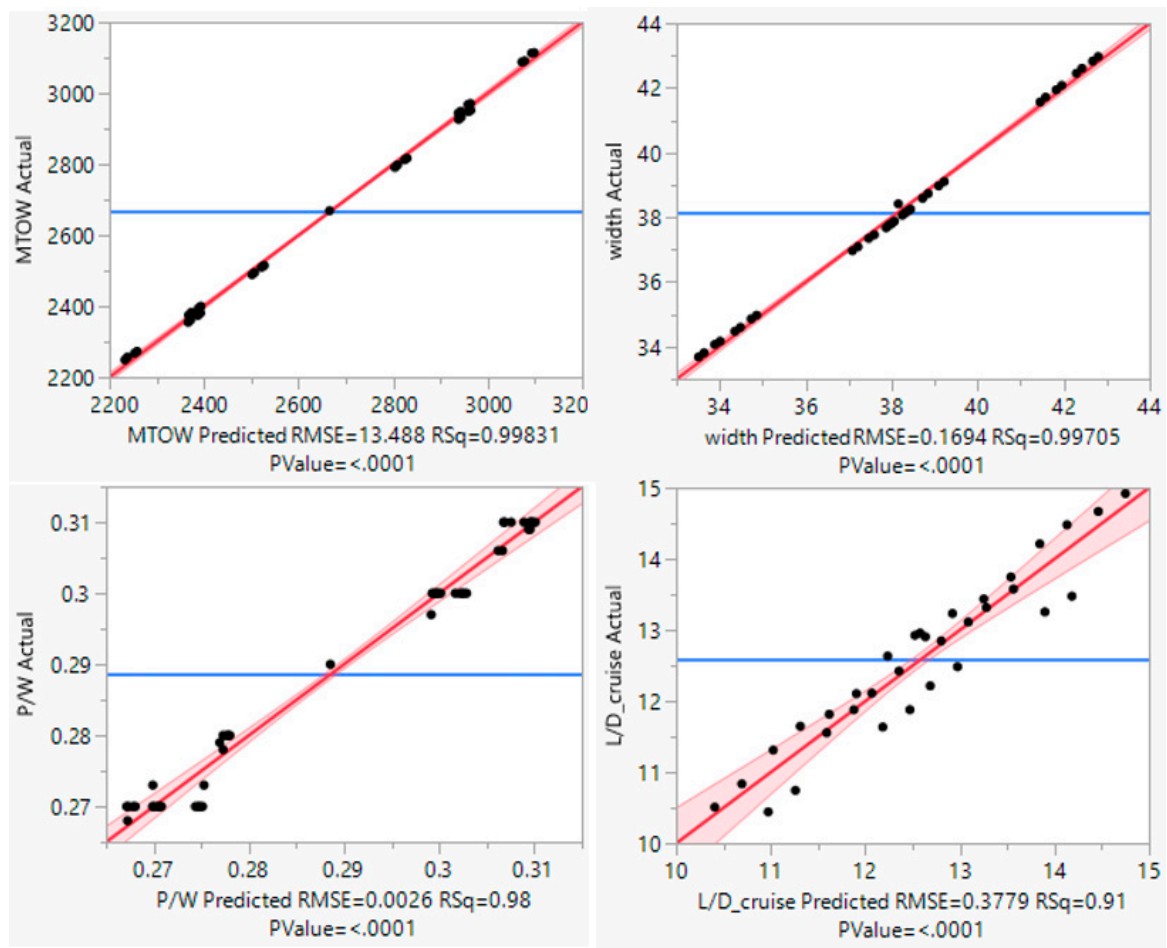

**Figure 12.** Screening test graph.

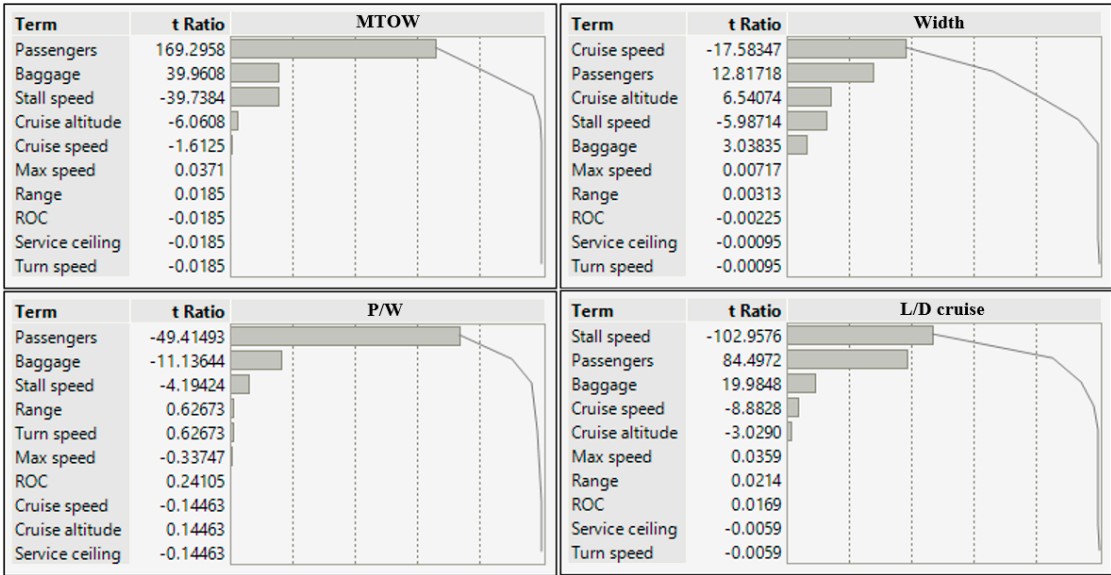

**Figure 13.** Screening test Pareto plot.

The regression coefficient, which is the result of the regression analysis, represents the relationship between the target design parameter and the independent design parameter. The *p*-value shown in Table 15 is used to determine whether the regression coefficient representing this relationship is an

appropriate value. The *p*-value is expressed as a value between 0 and 1, and the closer to 0, the better the relationship. In general, the null hypothesis is considered rejected when the significance level has a value of 0.05 or less, and represents the relationship between the target design parameter and the independent design parameter well.

**Table 15.** Independent design parameter *p*-value.

| Parameter | MTOW | P/W | Span | L/D Cruise |
|---|---|---|---|---|
| Intercept | <0.0001 | <0.0001 | <0.0001 | <0.0001 |
| Range | 0.9853 | 0.5335 | 0.9975 | 0.9830 |
| Max speed | 0.9706 | 0.7371 | 0.9943 | 0.9715 |
| **Cruise speed** | 0.1127 | 0.8855 | <0.0001 | <0.0001 |
| **Cruise altitude** | <0.0001 | 0.8855 | <0.0001 | 0.0038 |
| **Passengers** | <0.0001 | <0.0001 | <0.0001 | <0.0001 |
| **Baggage** | <0.0001 | <0.0001 | 0.0037 | <0.0001 |
| Rate of climb | 0.9853 | 0.8104 | 0.9982 | 0.9866 |
| **Stall speed** | <0.0001 | <0.0001 | <0.0001 | <0.0001 |
| Service ceiling | 0.9853 | 0.8855 | 0.9992 | 0.9953 |
| Turn speed | 0.9853 | 0.5335 | 0.9992 | 0.9953 |

In the present study, using the *p*-value of the regression coefficients and the results obtained from the regression analysis, five design independent parameters that greatly affect the design target parameters were selected, which are cruise speed, cruise altitude, passengers, baggage, and stall speed.

- Response surface method

In the response surface method (RSM), which is a statistical analysis method, when constructing a model in which design independent parameters have multiple effects on target design parameters, the response surface of the independent design parameters is estimated using a regression equation to optimize the model with minimum experiments.

In this paper, the following equation, which is the central composite design (CCD) method among the response surface methods, was used as the second regression model [22].

$$y = \beta_0 + \sum_{i=1}^{k} \beta_i x_i + \sum_{i=1}^{k} \beta_{ii} x_i^2 + \sum_{i=1}^{k-1} \sum_{j=i+1}^{k} \beta_{ij} x_i x_j, \tag{23}$$

where $y$ is the response, $x_i$ and $x_j$ are the design parameters, and $\beta_0$, $\beta_i$, $\beta_{ii}$, and $\beta_{ij}$ are the regression coefficients.

As independent design parameters, five factors selected through screening tests were used. The CCD was used in the JMP statistical software and conducted 43 experiments. The graph of the result of the response surface method obtained using the central composite design is shown in Figure 14. The narrower the graph, the better the modeling.

The coefficient of determination ($R^2$) and residual are the indices used to determine whether the response surface modeled through the central composite design is appropriate. The coefficient of determination is used as an indicator when evaluating the fitness of a regression model for appropriateness, or determining the relationship between a target design parameter and an independent design parameter. The formula of the coefficient of determination is as follows [23].

$$R^2 = 1 - \frac{\sum \left(Y_j - \hat{Y}_j\right)^2}{\sum \left(Y_j - \overline{Y}\right)^2} \tag{24}$$

where $Y_j$ is the exact response, $\hat{Y}_j$ is the predicted response, and $\overline{Y}$ is the mean value of the exact response.

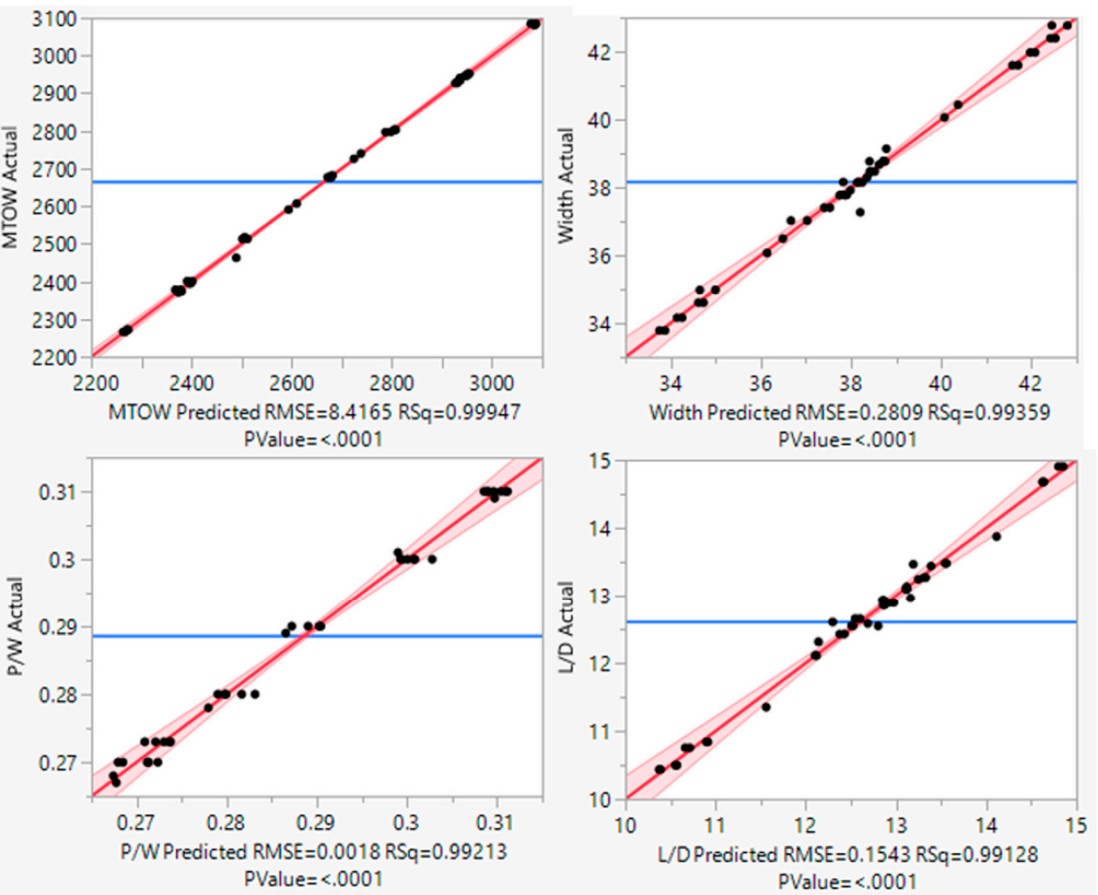

**Figure 14.** Response surface method graph.

However, in general, the coefficient of determination tends to increase as the number of design independent parameters increases, so it may not be an appropriate standard. Therefore, it is necessary to introduce a new indicator, the adjusted coefficient of determination ($AR^2$). The adjusted coefficient can be supplemented by considering the size of the sample and the number of independent design parameters. The equation of the adjusted coefficient of determination is as follows [23].

$$AR^2 = 1 - \frac{\left(1 - R^2\right)(n - 1)}{(n - k - 1)} = \frac{(n - 1)R^2 - k}{n - k - 1} \tag{25}$$

where $n$ is the sample size and $k$ is the number of independent design parameters.

The adjusted coefficient of determination has a value between 0 and 1, and it can be judged that the closer to 1, the better the relationship between the target design parameter and the independent design parameter. The coefficient of determination and the adjusted coefficient of determination are shown in Table 16 for a Vahana type vehicle. The graph of the residual is shown in Figure 15. In the residual graph, if the results are clustered or if there is a specific pattern, it can be considered a poor-fit.

**Table 16.** $R^2$, $AR^2$ of response surface method.

| Objective Design Parameter | $R^2$ | $AR^2$ |
|:---:|:---:|:---:|
| MTOW | 0.99 | 0.99 |
| P/W | 0.99 | 0.99 |
| Span | 0.99 | 0.99 |
| L/D cruise | 0.99 | 0.99 |

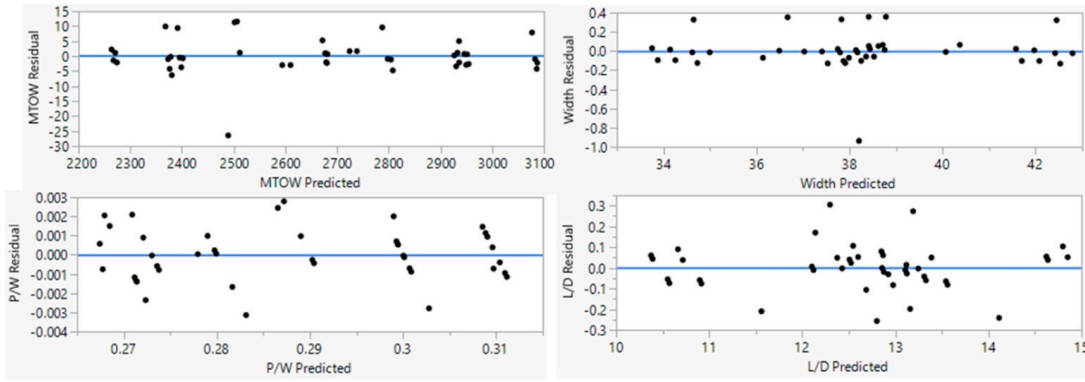

**Figure 15.** Response surface method residual graph.

- Monte Carlo simulation

A Monte Carlo simulation was used to validate if there was any feasibility when applying the model obtained using the response surface method. By generating 10,000 random numbers within the design range of the independent design parameters constituting the model, a combination of the same number of design parameters was generated, and DOEs were conducted to find the predicted values of the model. From the predicted value, the feasibility was determined according to the range of design target parameters. The corresponding values are summarized in Tables 17–20.

**Table 17.** Maximum takeoff weight (MTOW) percentile.

| Quantiles | | Predicted Value [lb] | Objective Value [lb] |
|---|---|---|---|
| 100.0% | Maximum | 2740 | |
| 75.0% | Quartile | 2487 | |
| 50.0% | Median | 2355 | |
| 25.0% | Quartile | 2214 | ≤2200 |
| 22.2% | | 2200 | |
| 0.0% | Minimum | 1989 | |

**Table 18.** Horsepower to weight ratio (P/W) percentile.

| Quantiles | | Predicted Value [hp/lb] | Objective Value [hp/lb] |
|---|---|---|---|
| 100.0% | Maximum | 0.3009 | |
| 75.0% | Quartile | 0.2855 | |
| 50.0% | Median | 0.2778 | |
| 26.2% | | 0.2700 | ≤0.27 |
| 25.0% | Quartile | 0.2696 | |
| 0.0% | Minimum | 0.2538 | |

**Table 19.** Width percentile.

| Quantiles | | Predicted Value [ft] | Objective Value [ft] |
|---|---|---|---|
| 100.0% | | 50 | |
| 100.0% | Maximum | 38 | |
| 75.0% | Quartile | 35 | |
| 50.0% | Median | 34 | ≤50 |
| 25.0% | Quartile | 33 | |
| 0.0% | Minimum | 30 | |

**Table 20.** L/D cruise percentile.

| Quantiles | | Predicted Value [-] | Objective Value [-] |
|---|---|---|---|
| 100.0% | Maximum | 14.0 | |
| 75.0% | Quartile | 12.6 | |
| 50.0% | Median | 12.1 | |
| 41.9% | | 12.0 | ≥12 |
| 25.0% | Quartile | 11.6 | |
| 0.0% | Minimum | 10.0 | |

- Design parameter optimization

In order to optimize the model obtained using the response surface method, a satisfaction function provided by JMP software was used. The maximum target takeoff weight, power to weight ratio, and span were set as the minimum value characteristics, and the lift to drag ratio was set as the maximum value characteristics, and optimization was performed. The design parameters optimized using the satisfaction function are summarized in Table 21. Also, the target design parameters were calculated using the optimized design parameters, and are shown in Table 22.

**Table 21.** Desirability function.

| Independent Design Parameter | Desirability | Optimized Value |
|---|---|---|
| Cruise speed | −0.095 | 109.05 [kt] |
| Cruise altitude | 0.079 | 7697.5 [ft] |
| Passengers | 1 | 3 [PAX] |
| Baggage | 1 | 120 [lb] |
| Stall speed | 1 | 40 [kt] |

**Table 22.** Optimized design parameter.

| Objective Design Parameter | Objective Value | Optimized Value |
|---|---|---|
| MTOW | ≤2200 [lb] | 2484 [lb] |
| P/W | ≤0.27 [hp/lb] | 0.25 [hp/lb] |
| Width | ≤50 [ft] | 37 [ft] |
| L/D cruise | ≥12 | 12 |

## 4. Conclusions

Using design of experiments, this study conducted an optimization analysis of the design parameters and conceptual design of an eVTOL PAV, a new concept aircraft to be used for urban air mobility (UAM).

In the defining stage of the problem, the design goal of the eVTOL PAV was determined, and the maximum takeoff weight, power to weight ratio, span, and lift to drag ratio were set as design target parameters to meet the design requirements. For the design goal setting and detailed settings of the eVTOL PAV, data provided by Uber and Vahana were used.

In order to derive an optimized model that satisfied the design target parameters, a comparative analysis study combination table was constructed and a combination of numerous subsystems constituting the eVTOL PAV was constructed to effectively set the reference model. The model was based on the Vahana tilted wing type.

In order to find a model that satisfied the designated design target parameters, the physical design parameters that determine the performance of eVTOL PAV were set as design independent parameters. A new model was designed using independent parameters with high sensitivity to design target parameters, through screening tests among independent design parameters. The model was designed

using the central composite design method among the response surface method, and a Monte Carlo simulation was applied to determine the design feasibility of the model.

As a result of the Monte Carlo simulation, the design target parameter showed a potential design with a maximum takeoff weight of 22.2%, a power to weight ratio of 26.6%, a span of 100%, and a lift to drag ratio of 41.9%. In addition, the design independent parameter was optimized using the satisfaction function provided by JMP software, and the optimized design target parameters were derived using the optimized design independent parameters. Among the optimized design target parameters, the remaining design target parameters, excluding the maximum takeoff weight, reached the design target value.

Many companies are currently participating in eVTOL PAV development. However, the eVTOL PAV is still in the research phase, and so few companies have disclosed aircraft data. This has made it difficult to employ aircraft concept design techniques using traditional statistical data for the conceptual design of the eVTOL PAV.

Accordingly, this paper proposed a method for conducting an eVTOL PAV concept design effectively in the absence of existing design data, and for identifying alternatives applicable to the eVTOL PAV concept design phase, using a self-developed PAV sizing program and JMP software. In addition, the sensitivity of the design parameters needed to satisfy the design target values was analyzed, and optimization and design possibilities were presented.

**Author Contributions:** B.-S.L. was responsible for all tasks related to the work, from establishing the methodology, and all data analysis; A.T. reviewed the original manuscript; H.-Y.H. was responsible for conceptualization, the writing of the original manuscript, and managed the overall project and reviewed it. All authors have read and agreed to the published version of the manuscript.

**Funding:** This research was supported by the Research Grant from Sejong University through the Korea Agency for Infrastructure Technology Advancement funded by the Ministry of Land, Infrastructure and Transport of the Korean government (Project No.: 20CTAP-C157731-01).

**Acknowledgments:** This research was supported by the Research Grant from Sejong University through the Korea Agency for Infrastructure Technology Advancement funded by the Ministry of Land, Infrastructure and Transport of the Korean government (Project No.: 20CTAP-C157731-01).

**Conflicts of Interest:** The authors declare no conflict of interest.

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
