# Peer review of "Optimal Design and Design Parameter Sensitivity Analyses of an eVTOL PAV in the Conceptual Design Phase"

_applsci, doi:10.3390/app10155112_

Round 1

Reviewer 1 Report

Following the reading of the paper, believe that this paper should be published after its minor revisions.

The suggested minor revisions are the following:

1) there is the need to add a ref. where is needed, for example, to explain certain equations (even classical equations)

2) the explanation of some equations needs improvement, as for example: 

2.1) replace ''Summarizing Equations (6) through (9)'' by ''Replacing CL defined in Eq. (8) into Eq. (6)''

2.2) replace ''Using Equation (10)'' by ''Replacing CDi defined in Eq. (10) into Eq. (5)''

3) First phrase of the section 3.1 should be modified into:

''In general, the procedure for a DOE is explained in Figure 8'' instead ''In general, the procedure for a DOE is as shown in Figure 8''. 

Many thanks in advance for the publication of this paper after its minor corrections. 

Reviewer 2 Report

This paper presents an optimisation study for the conceptual design of an eVTOL PAV. The paper is well written and contains interesting results. Some problems must be addressed before it is suitable for publication, these problems being related mostly to the level of detail provided for certain aspects of the methodology.

Comments:

1) The literature review should be expanded to include works on the topic of eVTOL conceptual design studies, and highlight the design strategies used therein.

2) The authors should justify at the beginning of section 2.1 why they chose the tilt-wing Vahana configuration as the starting point for their study.

3) There is a very large difference in the CL values of the main wing between XFRL5 and OpenVSP. Could it be due to the lack of wingtip surfaces in XFLR5? The authors should comment on this and indicate which set of results they used for the rest of the study. Also, all three lifting surfaces should be included in both XFLR5 and OpenVSP results, not just two.

4) The Vahana design has several propulsion units distributed along the wing span. These will disturb the spanwise lift distribution compared to the clean wing analysed. Can the authors quantify the magnitude of these changes?

5) How were the results in Figure 7 obtain? I am guessing it is somehow based on the cruise values given in Table 2, but the authors must indicated how these were calculated.

6) On page 14, the authors should describe in more detail the methods used in the weight, aerodynamic, propulsion sheets, etc. This is especially important since only the Aerodynamics was detailed in the paper, but hardly anything was mentioned on the other elements.
